Improved cartilage regeneration by implantation of acellular biomaterials after bone marrow stimulation: a systematic review and meta-analysis of animal studies

Pot Michiel W. 1
Gonzales Veronica K. 2
Buma Pieter 2
IntHout Joanna 3
van Kuppevelt Toin H. 1
de Vries Rob B.M. 4
Daamen Willeke F. Willeke.Daamen@radboudumc.nl 1
1 Department of Biochemistry, Radboud Institute for Molecular Life Sciences, Radboud university medical center , Nijmegen , The Netherlands
2 Department of Orthopedics, Radboud Institute for Molecular Life Sciences, Radboud university medical center , Nijmegen , The Netherlands
3 Department for Health Evidence, Radboud Institute for Health Sciences, Radboud university medical center , Nijmegen , The Netherlands
4 SYRCLE (SYstematic Review Centre for Laboratory animal Experimentation), Central Animal Laboratory, Radboud university medical center , Nijmegen , The Netherlands
Gentleman Eileen
Electronic publication date: 2016 Sep 8
Publication date: 2016
Volume: 4
Electronic Location ID: e2243
Received 2016 Feb 26; Accepted 2016 Jun 21
Copyright: ©2016 Pot et al.
Copyright year: 2016
Copyright holder: Pot et al.
License: This is an open access article distributed under the terms of the Creative Commons Attribution License, which permits unrestricted use, distribution, reproduction and adaptation in any medium and for any purpose provided that it is properly attributed. For attribution, the original author(s), title, publication source (PeerJ) and either DOI or URL of the article must be cited.
License URL: https://creativecommons.org/licenses/by/4.0/

Keywords: Regenerative medicine, Cartilage, Microfracture, Osteochondral, Biomaterials, Scaffold, Cell-free

Funding: Dutch government FES0908 Netherlands Organisation for Health Research and Development 104024065 This work was supported by a grant from the Dutch government to the Netherlands Institute for Regenerative Medicine (NIRM, grant No. FES0908). Rob de Vries received funding from The Netherlands Organisation for Health Research and Development (ZonMw; grant nr. 104024065). The sources of funding have no other involvement in this publication. The funders had no role in study design, data collection and analysis, decision to publish, or preparation of the manuscript.

==============================
Microfracture surgery may be applied to treat cartilage defects. During the procedure the subchondral bone is penetrated, allowing bone marrow-derived mesenchymal stem cells to migrate towards the defect site and form new cartilage tissue. Microfracture surgery generally results in the formation of mechanically inferior fibrocartilage. As a result, this technique offers only temporary clinical improvement. Tissue engineering and regenerative medicine may improve the outcome of microfracture surgery. Filling the subchondral defect with a biomaterial may provide a template for the formation of new hyaline cartilage tissue. In this study, a systematic review and meta-analysis were performed to assess the current evidence for the efficacy of cartilage regeneration in preclinical models using acellular biomaterials implanted after marrow stimulating techniques (microfracturing and subchondral drilling) compared to the natural healing response of defects. The review aims to provide new insights into the most effective biomaterials, to provide an overview of currently existing knowledge, and to identify potential lacunae in current studies to direct future research. A comprehensive search was systematically performed in PubMed and EMBASE (via OvidSP) using search terms related to tissue engineering, cartilage and animals. Primary studies in which acellular biomaterials were implanted in osteochondral defects in the knee or ankle joint in healthy animals were included and study characteristics tabulated (283 studies out of 6,688 studies found). For studies comparing non-treated empty defects to defects containing implanted biomaterials and using semi-quantitative histology as outcome measure, the risk of bias (135 studies) was assessed and outcome data were collected for meta-analysis (151 studies). Random-effects meta-analyses were performed, using cartilage regeneration as outcome measure on an absolute 0–100% scale. Implantation of acellular biomaterials significantly improved cartilage regeneration by 15.6% compared to non-treated empty defect controls. The addition of biologics to biomaterials significantly improved cartilage regeneration by 7.6% compared to control biomaterials. No significant differences were found between biomaterials from natural or synthetic origin or between scaffolds, hydrogels and blends. No noticeable differences were found in outcome between animal models. The risk of bias assessment indicated poor reporting for the majority of studies, impeding an assessment of the actual risk of bias. In conclusion, implantation of biomaterials in osteochondral defects improves cartilage regeneration compared to natural healing, which is further improved by the incorporation of biologics.

Introduction

Articular cartilage is a specialized tissue that covers joint surfaces and provides a low-friction and load-bearing surface for a smooth motion of joints. The structure and function of the tissue can be compromised by traumatic injuries and degenerative joint diseases. Due to its avascular nature, damaged cartilage tissue does not heal spontaneously and it remains a challenge to fully restore tissue function (Ahn et al., 2009; Cao et al., 2012).

The surgical options to treat patients with a localized cartilage defect are limited to cartilage regeneration approaches such as autologous chondrocyte implantation and microfracture surgery (Aulin et al., 2013; Bal et al., 2010). The latter strategy, also known as bone marrow stimulation, is relatively simple, minimally invasive and inexpensive. During this procedure the subchondral bone plate below the cartilage lesion is perforated to initiate bleeding and induce a reparative response. The principle behind this regenerative resurfacing strategy is the migration of non-differentiated bone marrow-derived multipotent stem cells from the subchondral bone into the defect site leading to the formation of new cartilage tissue (Buma et al., 2003; De Mulder et al., 2014; Erggelet et al., 2009). Patients treated with bone marrow stimulation generally show clinical improvements up to 1.5–3 years after surgery. However, five years after surgery higher incidences of clinical failures are observed (Hoemann et al., 2010; Van der Linden et al., 2013). The newly formed tissue generally consists of fibrocartilage repair tissue rather than hyaline cartilage, has limited filling of the defect, integrates poorly with the surrounding tissue and has inferior mechanical properties compared to hyaline cartilage (Dai et al., 2014). Therefore, the need for regeneration of more durable cartilage tissue persists.

Regenerative medicine and tissue engineering may offer promising alternatives and/or additions to clinical strategies that aim to restore damaged cartilage tissue. The construction of biomaterials and the incorporation of cells and biologics in these implants have been widely investigated for this purpose. Biomaterials can be implanted in osteochondral defects created by applying marrow stimulating techniques (microfracture and subchondral drilling (Falah et al., 2010)) to guide and stimulate the formation of cartilage tissue (Seo et al., 2014). During microfracture surgery, an arthroscopic awl is used to penetrate the subchondral bone, while with subchondral drilling a high speed drill is applied to penetrate the trabecular bone. Different strategies have been applied including the implantation of biomaterials with and without cells. Acellular biomaterials offer various advantageous properties such as lack of donor-site morbidity, absence of cell culture costs, off the shelf availability, fewer regulatory issues, and application of one-stage surgical procedures (Brouwer et al., 2011; Efe et al., 2012). Many researchers have explored the approach of implanting acellular biomaterials and investigated the use of various biomaterials in vivo, such as natural (e.g., collagen (Breinan et al., 2000; Buma et al., 2003; Enea et al., 2013; Wakitani et al., 1994), chitosan (Abarrategi et al., 2010; Bell et al., 2013; Guzman-Morales et al., 2014; Hoemann et al., 2007), alginate (Igarashi et al., 2012; Mierisch et al., 2002; Sukegawa et al., 2012) and hyaluronic acid (Aulin et al., 2013; Kayakabe et al., 2006; Marmotti et al., 2012; Solchaga et al., 2000)) and synthetic polymers (e.g., polycaprolactone (Christensen et al., 2012; Martinez-Diaz et al., 2010; Mrosek et al., 2010), polyvinyl alcohol (Coburn et al., 2012; Holmes, Volz & Chvapil, 1975; Krych et al., 2013) and poly(lactic-co-glycolic acid) (Athanasiou, Korvick & Schenck Jr, 1997; Chang et al., 2012; Cui, Wu & Hu, 2009; Fonseca et al., 2014)). To combine the advantageous properties of these materials, multilayered biomaterials (e.g., β-tricalcium phosphate-hydroxyapatite/hyaluronate-atelocollagen (Ahn et al., 2009), ceramic bovine bone-gelatin/gelatin-chondroitin sulfate-sodium hyaluronate (Deng et al., 2012)), blends (e.g., poly(glycolic acid)-hyaluronic acid (Erggelet et al., 2009) and type I collagen-hyaluronic acid-fibrinogen hydrogel (Lee et al., 2012)) have been constructed. Biologics are natural factors that can be used to stimulate tissue regeneration, e.g., by inducing proliferation and differentiation of cells. Biologics such as growth factors of the transforming growth factor β (TGF-β) superfamily and others have been incorporated in biomaterials to guide and stimulate the formation of hyaline cartilage tissue (Richter, 2009). Moreover, it has been reported that the animal model of choice may have a significant impact on study outcome of articular cartilage regeneration (Reinholz et al., 2004). Currently, there is no systematic overview of the current literature assessing the effect of various parameters (e.g., applied biomaterials, incorporated biologics and animal models) on cartilage regeneration.

The aim of this systematic review and meta-analysis is to assess all current evidence for the efficacy of articular cartilage regeneration using acellular biomaterials implanted in the knee and ankle joint after microfracture and subchondral drilling in animal models. Additionally, we strive to provide transparency on the quality of performed in vivo studies, in order to aid the design of future animal experiments and clinical trials. We provide a systematic and unbiased overview of the current literature addressing regeneration of articular cartilage using a wide range of acellular biomaterials containing various biological cues (as illustrated in Fig. 1). Results of semi-quantitative histological scoring systems are used as a quantitative outcome parameter for outcome assessment of cartilage regeneration. Although microfracture surgery and subchondral drilling strive to stimulate cartilage and osteochondral regeneration, respectively, both are generalized in this study as cartilage regeneration. Moreover, the evaluation of different subgroups (natural and synthetic origin of the biomaterials, structure of the materials (scaffolds vs. hydrogels), incorporated biological cues, and animal models) was included to gain insights in which parameters affect cartilage regeneration and to what extent.

Figure 1 Illustration of cartilage regeneration by implantation of biomaterials after bone marrow stimulation.

The implanted biomaterials provide a template to guide cartilage regeneration by bone marrow derived mesenchymal stem cells.

Materials and Methods

Search strategy

To identify relevant peer-reviewed articles, a comprehensive search of the literature using PubMed and EMBASE (via OvidSP) was conducted, using the methods defined by De Vries et al. (2012) and Leenaars et al. (2012). The last search date was April 3rd 2015. In both databases, a tissue engineering search component developed by Sloff et al. (2014), consisting of equivalents for tissue engineering (e.g., tissue regeneration, regenerative medicine, bio-engineering or biomatrices), was combined with a cartilage search component, consisting of equivalents for cartilage and cartilage-related surgeries (e.g., chondral, chondrogenic, surgery, microfracturing or implants). The search components were constructed using MeSH terms (PubMed) and EMTREE terms (EMBASE) and additional free-text words from titles or abstracts ([tiab] or ti,ab). The obtained tissue engineering-related and cartilage-related results were filtered for animal studies using previously described animal search filters (De Vries et al., 2011; Hooijmans et al., 2010). The complete search strategy is attached in Supplemental Information 1. No language restrictions were used.

Study selection

References from the PubMed and EMBASE search strategies were combined and duplicates were manually removed from EndNote, with the preference of PubMed over EMBASE. All screening phases were performed by two independent reviewers (MP and VG) and reported according to the “Preferred Reporting Items for Systematic Reviews and Meta-Analysis” (PRISMA) guidelines (Higgins & Green, 2011). References were first screened based on title and were excluded based on the following criteria: (1) titles showed no relevance to regeneration of articular (hyaline) cartilage, (2) it was specifically stated in the title that the conducted experiment was an in vitro study only, (3) osteoarthritis animal models were used, (4) only ex vivo studies were performed, and (5) deceased animals were used. In case of doubt or disagreement, references were included for further screening. The second screening phase consisted of a title/abstract screening in Early Review Organizing Software (EROS, Institute of Clinical Effectiveness and Health Policy, Buenos Aires, Argentina; www.eros-systematic-review.org). References were included based on the following inclusion criteria: (1) primary study, (2) animal model, (3) bone marrow stimulation by microfracturing or creation of an osteochondral defect, and (4) biomaterial implantation. Articles were only excluded when it was specifically stated in the abstract that the study was performed without healthy animals or acellular biomaterials, or if biomaterials were not implanted in the knee or ankle joint. Articles were not excluded in case important information in the abstract was missing. These articles were assessed in the full-text screening phase. For the full-text screening, articles were included if they met all of the following inclusion criteria: (1) primary study, (2) animal model, (3) healthy animals, (4) articular cartilage regeneration, (5) knee or ankle joint, (6) bone marrow stimulation by microfracturing or creation of an osteochondral defect, and (7) implantation of an acellular biomaterial. In general, if results of the two reviewers were different, articles were discussed until consensus was reached. In case of double publication, one of the studies was removed. During the screening phase, no selection was made based on publication language. The risk of bias assessment and meta-analysis was applied to studies with a comparison between a non-treated empty defect control and biomaterial implantation, and with semi-quantitative histological scoring system results as outcome data.

Study characteristics

From the studies included after the full-text screening, the following details were obtained: general information (author and year of publication), animal characteristics (species, strain, sex, age, weight and the number of animals), information related to the surgical defect (size, depth and location), experimental conditions, biomaterial, biologics, evaluation time points and all outcome measures used, i.e., macroscopic evaluation, semi-quantitative macroscopic evaluation, histology, immunohistochemistry, semi-quantitative histological scoring, and biomechanical tests. Data from semi-quantitative histological scorings were used in the meta-analysis (described in ‘Analysis preparations and meta-analysis’). Histological scoring systems applied in different studies consisted of scoring parameters like cell morphology, Safranin-O staining, integrity of surface, thickness, surface of area filled with cells, chondrocyte clustering, degenerative changes, restoration of the subchondral bone and integrity.

Risk of bias assessment

A risk of bias analysis was performed to assess the methodological quality of the studies included in the meta-analysis, using an adapted version of the risk of bias tool described by Hooijmans et al. (2014) (for all included studies containing a ‘non-treated empty defect’ as control group and studies using semi-quantitative histological scoring systems as outcome measure). A flowchart was constructed (Supplemental Information 2) to score for selection, performance, detection and attrition bias, where the scores ‘−’, ‘?’ and ‘+’ indicate a low, unknown and high risk of bias, respectively. The questions addressed are specified in the Supplemental Information 2. Articles were scored independently by MP and VG, and if the results of the two reviewers were different, results were discussed until consensus was reached. All articles written in Chinese (16 studies) were excluded from the risk of bias assessment only, due to limited resources to independently translate these articles by two native Chinese speakers. However, the data of these studies were extracted and used in the meta-analysis.

Analysis preparations and meta-analysis

Analysis preparations

The statistical analyses were restricted to those studies containing the outcome measure semi-quantitative histology, making a comparison between a ‘non-treated empty defect’ as control group and implanted biomaterials as experimental group. Data (mean, standard deviation (SD) and number of animals) of the control and experimental group were extracted from the studies, for all available time points. When results were not given numerically, but depicted graphically, the mean and SD were measured using ImageJ (1.46r, National Institutes of Health USA). For studies presenting results in boxplots, the mean and standard deviation were recalculated from the median, range and the sample size according the method described by Hozo, Djulbegovic & Hozo (2005). When data were described by a mean and confidence interval (CI), the CI was recalculated to a standard deviation by the following equation: standard deviation=N×upper limit−lower limit3.92 for a 95% CI (Higgins & Green, 2011). For some studies, data were unclear and assumptions were made, which are listed in Supplemental Information 3. To compare studies with different histological score system scales, means and standard deviations were converted to a 100% scale by dividing the result by the maximum achievable histological score and multiplying by 100%. In case of missing or unclear data, authors were e-mailed to retrieve the data. When data could not be obtained, these studies were excluded from the meta-analysis (reasons for exclusion are also given in Supplemental Information 3). Results of studies with several experimental groups were combined, following the approach described in the Cochrane Handbook, table 7.7 (Higgins & Green, 2011). The same approach was followed to combine results of different animals on several time points in the same group in the same study. One study (Hamanishi et al., 2013) had an SD of zero, which caused problems in the analyses. Therefore, the SD was changed to 4.29, equal to the SD of the experimental group of the same study at the same time point. The resulting data were used to calculate the treatment effect and corresponding standard error (SE) per study.

Meta-analysis

The following main research question was assessed: Does an overall beneficial effect exist of implanting acellular biomaterials in osteochondral defects compared to non-treated empty defects?

First, in order to select the appropriate statistical random-effects meta-analysis model, we compared a univariate approach to the bivariate approach. In the bivariate approach, separate outcomes for control and experimental group were used with their respective SEs. The correlation between these two outcomes was modeled with a compound symmetry covariance matrix, as this resulted in a much lower Akaike Information Criterion value than the use of an unstructured covariance matrix. Results were compared with those of the univariate approach, based on the treatment effect and SE per study. Results of the univariate and bivariate approaches were very similar and we therefore proceeded with the univariate approach, when applicable in combination with likelihood ratio tests.

Restricted to the experimental groups, the following sub-questions were addressed to evaluate whether the treatment effect depended on specific variables: (1) Is there a difference between the use of natural and synthetic biomaterials?; (2) Does the structure of the biomaterials affect cartilage regeneration?; (3) Do differences among various material subgroups exist?; (4) Does incorporation of biologics have a beneficial effect on cartilage regeneration compared to control biomaterials?; (5) Do differences among subgroups of biologics exist?; (6) Do different animal models result in variations in cartilage regeneration? Results are shown as % cartilage regeneration (95% CI: [lower CI, upper CI]. Some studies have more than one experimental group. Therefore, the total number of studies and number of experimental groups (no. of studies/groups) are provided.

Sensitivity analyses were performed to evaluate the effect of time (e.g., all time points, short (≤8 weeks), long time points (>8 weeks), or the maximum time point), outliers (excluding consecutively the studies with the 10% highest/lowest pooled SD, and studies with the 10% highest/lowest SE), implant location, bone marrow stimulating technique applied (microfracturing vs. subchondral drilling), language (excluding studies reported in Chinese as the risk of bias of these studies was not assessed), and excluding studies where assumptions had to be made. Based on a pilot analysis, data of all time points were used for subgroup analyses. Subgroup analyses were only performed for subgroups consisting of more than two groups.

The statistical analyses were performed with SAS/STAT® software version 9.2 for Windows (SAS Institute Inc., Cary, NC, USA). The funnel plot shows the overall outcome of the pooled effect size of each study. I2 was used as a measure of heterogeneity. The forest plot was created with ReviewManager (RevMan, Version 5.3, 2014; The Cochrane Collaboration, The Nordic Cochrane Centre, Copenhagen, Denmark).

Results

Search and study inclusion

The searches conducted in PubMed and EMBASE (Supplemental Information 1) resulted in 4,401 and 5,986 studies, respectively, leaving 6,688 studies after removal of duplicates. These studies were screened by title and title/abstract, which resulted in 1,088 included studies after the title screening and 517 included studies after the title/abstract screening. Screening articles by full-text and subsequently selection for studies with empty defect controls as well as semi-quantitative histology as outcome measure resulted in 283 included studies after full-text assessment, of which 151 and 135 articles could be used for the meta-analysis and risk of bias assessment, respectively (Fig. 2). The studies from Xie et al. (2014), Yao, Ma & Zhang (2000) and Zhou & Yu (2014) could not be retrieved as a full text and these studies were therefore excluded. An overview of all included studies after full-text assessment as well as studies included for the risk of bias assessment and meta-analysis is provided in Supplemental Information 3. All references and abbreviations can be found in Supplemental Information 4. In this table, remarks are provided related to exclusion reasons for risk of bias assessment and meta-analysis (e.g., duplicate publication and incomplete data). Assumptions made for certain studies are also stated in this table.

Figure 2 PRISMA (Preferred Reporting Items for Systematic Reviews and Meta-analysis) flowchart of the systematic search of literature.

Study characteristics

The study characteristics (Supplemental Information 3) clearly show substantial variation among studies. A wide range of animal species was used, from small (rat and rabbit) to larger animal models (dog, minipig, goat, pig, sheep and horse). A large variation was observed between the ages of animals (e.g., the age of rabbits ranged from 6 weeks to >2 years). Often ages were not described or specified specifically (e.g., as adult or mature). Generally, the animals were older (range of years) in large animal models compared to animals used in small animal models (range of months). The defects were created at different locations in the knee joint, such as the trochlea, condyle (medial and lateral), femur and intercondylar fossa. In addition, a large variation was found in the dimensions of the prepared defects, e.g., the dimensions of the defects created in rabbits ranged from 4–7 mm in diameter and 0.8–9 mm in depth. Microfracture surgery and subchondral drilling was performed in 25 and 258 studies, respectively. The implanted biomaterials were of natural or synthetic origin or combinations thereof, and consisted of single-layered or multilayered implants or blends thereof. Implants were constructed from a wide range of materials or combinations thereof, such as collagen, chitosan, hyaluronic acid, alginate, fibrin, hydroxyapatite, poly(lactic-co-glycolic acid), polycaprolactone, poly(glycolic acid) and poly(ethylene glycol), and used in different states: scaffolds, hydrogels, or hybrid mixtures of both. Various biological cues were incorporated in the biomaterials prior to implantation or administered afterwards by injection into the knee joint, mostly growth factors of the TGF-β superfamily such as bone morphogenetic protein 2 (BMP-2) and TGF-β1, but also fibroblast growth factor (FGF) and platelet-rich plasma (PRP). The maximum follow-up time was 1 year, but studies mainly investigated relatively short-term effects of implanted biomaterials on cartilage regeneration (up to 6 months).

Risk of bias assessment

A risk of bias assessment was performed to assess risks of bias (selection, performance bias, detection and attrition bias) in studies included for the meta-analysis (Fig. 3). An overview of all scores per individual study is provided in Supplemental Information 6.

Figure 3 Risk of bias of all included studies in the meta-analysis.

The green, orange and red colors depict the percentages of studies with low, unknown or high risk of bias of the total number of assessed studies. The risk of bias assessment indicated a general lack of details regarding the experimental setup, as indicated by the orange bars. The green bars represent a low risk of bias, mainly for the difference between groups at the moment of surgical intervention and addressing incomplete outcome data. High risk of bias was infrequently scored, as indicated by the red bars. Q4–Q6 are not depicted in the graph, but are described in Supplemental Information 6.

The risk of bias assessment showed that details with respect to the randomization method were not provided (Q1). It was often described that animals were randomized across different groups without describing the method of randomization, thereby limiting assessment of the adequacy of randomization and therefore the actual risk of selection bias. Another notable observation from the experimental designs studied was that only in a limited number of studies it was described that power calculations were performed, whereas sufficient power in animal experiments is a requirement for performing adequate studies. The actual power analyses were never provided in the studies. Due to a lack of information, it was also difficult to assess possible bias by differences in implantation sites (with differences in load-bearing conditions, Q2.1) and differences between groups related to the age, sex and weight of the animals at the start of the experiment (Q2.2). Generally, baseline characteristics of animals prior to implantation of biomaterials (e.g., some animals received additional surgery related to harvesting of cells for biomaterials combined with cells, Q2.3) were similar. When implanting biomaterials, no details were described on blinding different biomaterials (Q3). Blinding of the empty defect and biomaterial conditions should be performed to limit bias. However, blinding between the empty defect and biomaterial group is impossible in case only one biomaterial is implanted. More than half of the studies conducted blinded outcome assessment while performing the histological scoring, resulting in low risk of detection bias, whereas the other studies had an unknown risk (Q7). For most studies, no incomplete outcome data were described/found, resulting in low risk of attrition bias. For some studies, dropouts were described/found, resulting in differences between groups and high risk of bias (Q8). Overall, the risk of bias analysis generally revealed poor reporting of the experimental design for the majority of the studies, impeding an assessment of the actual risk of bias.

Data synthesis

For an overview of the meta-analysis and results obtained, see Table 1. The histological scores of defects implanted with biomaterials and non-treated empty defects are presented as a percentage on a 100% scale, where 0% and 100% indicate poor and perfect cartilage regeneration, respectively. Data are presented as the effect (%) with 95% CI.

Table 1 Overview of the meta-analysis results for the main research question assessing the overall beneficial effect of implanting acellular biomaterials in osteochondral defects compared to non-treated empty defects and sub-questions evaluating the effect of specific variables on the treatment effect.

The total number of studies and number of experimental groups included in the meta-analysis are shown (some studies have >1 experimental group, no. of studies/groups). The quality of cartilage regeneration is presented on a 100% scale, where 100% represents the maximum achievable histological score and thus the best cartilage regeneration. Implantation of biomaterials significantly improved cartilage regeneration compared to non-treated empty defects, which was further improved by the incorporation of biologics. No significant differences were found between natural and synthetic materials, between the various material subgroups, and between the biomaterial structures (hydrogels versus scaffolds versus blends), and between animal species.

Meta-analysis	No. of studies/groups	Subgroups	Cartilage regeneration (% (95% CI))	Mean difference (% (95% CI)) p-value	
1. Overall effect	127/400	Biomaterial	53.6 [50.7, 56.6]	15.6 [12.6, 18.6]	
127/247	Empty defect	38.1 [35.1, 41.0]	p < 0.0001	
2. Origin materials	76/222	Natural	53.0 [49.3, 56.6]	−0.73 [−6.5, 5.0]	
39/137	Synthetic	53.7 [48.8, 58.7]	p = 0.887	
3. Material subgroups	20/68	Collagen	49.5 [41.1, 57.8]	p = 0.804	
6/17	Chitosan	57.5 [40.8, 74.2]	
5/11	Hyaluronic acid	47.9 [31.7, 64.1]	
5/16	Alginate	63.0 [46.9, 79.00]	
3/10	Fibrin	55.3 [34.4, 76.3]	
5/11	Bone	51.2 [35.2, 67.2]	
15/52	PLGA	58.5 [49.0, 68.0]	
6/21	PAMPS-PDMAAm DN	47.9 [31.7, 64.1]	
4. Scaffold structure	78/258	Scaffolds	53.1 [49.5, 56.7]	p = 0.973	
41/127	Hydrogels	54.2 [49.4, 59.1]	
7/17	Blends	55.7 [42.0, 69.3]	
5. Biologicals	113/291	No biologicals	51.7 [48.6, 54.9]	7.56 [2.1, 13.0]	
35/109	Biologicals	59.3 [54.0, 64.6]	p = 0.007	
6. Biological cues	9/35	BMP	56.6 [−6.3, 119.6]	p = 0.780	
5/20	FGF	51.8 [−43.9, 147.4]	
8/14	PRP	55.9 [−20.9, 132.8]	
6/16	TGF	60.2 [−7.5, 128.0]	
7. Animal models	3/5	Dogs	ED: 31.9 [14.5, 49.4]; B: 50.6 [33.0, 68.2]	18.7 [−0.0, 37.3]	
5/13	Goats	ED: 58.5 [43.4, 73.7]; B: 61.6 [47.6, 75.6]	3.1 [−13.2, 19.4]	
1/3	Macaques	ED: 12.2 [−18.2, 42.6]; B: 6.8 [−23.3, 37.0]	−5.4 [−37.6, 26.8]	
10/20	Minipigs	ED: 42.4 [32.4, 52.4]; B: 56.1 [46.3, 66.0]	13.6 [3.1, 24.1]	
94/333	Rabbits	ED: 37.7 [34.2, 41.1]; B: 52.5 [49.0, 55.9]	14.8 [11.1, 18.5]	
13/23	Sheep	ED: 35.3 [26.4, 44.3]; B: 61.3 [52.7, 70.0]	26.0 [16.3, 35.7]	
			p = 0.348	
Notes.

ED Empty defect

B Biomaterials

Overall effect biomaterial implantation

The meta-analysis indicates a significant improvement of cartilage regeneration using acellular biomaterials implanted after applying marrow stimulating techniques compared to non-treated empty defects (15.6% (95% CI [12.6, 18.6], p < 0.0001). The forest plot (Supplemental Information 7) depicts the outcome effect of each individual study. In 73 studies cartilage regeneration significantly improved by the incorporation of biomaterials. In 48 studies no effect was found, whereas in only six studies a negative effect on cartilage regeneration was observed. A similar significant effect was observed taking into account the maximum follow-up only (16.3% [13.1, 19.6], p < 0.0001). Also for short and long term follow-up cartilage regeneration was significantly improved (≤8 weeks: 12.5% [9.3, 15.7], >8 weeks: 17.1% [13.9, 20.2]). No notable differences in cartilage regeneration were found between the results based on the maximum follow-up time per study versus those based on all time points per study. Therefore, further subgroup analyses were made using results from all time points together.

Natural and synthetic materials

The subgroup analysis assessing cartilage regeneration using materials of different origin, natural and synthetic, indicated no significant differences (p = 0.887) between natural (53.0% [49.31, 56.63]) and synthetic materials (53.7% [48.75, 58.65]).

Dividing the group of materials into subgroups allows comparison of cartilage regeneration using different biomaterials. The following subgroups were studied: (1) collagen, (2) chitosan, (3) hyaluronic acid–based biomaterials), (4) alginate, (5) fibrin), (6) bone material-based, (7) PLGA, and (8) PAMPS-PDMAAm DN hydrogel. No significant differences between the biomaterial subgroups were found (Table 1).

Material structure

Materials were divided in three groups based on their structure: (1) scaffolds, (2) hydrogels, and (3) blends. Cartilage regeneration was similar after use of scaffolds (53.1% [49.53, 56.74]), hydrogels (54.2% [49.39, 59.07]) and blends (55.7% [42.0, 69.3), p = 0.973.

Biologics

Incorporation of biologics in the biomaterials resulted in a statistically significant improvement in cartilage regeneration of 7.6% [2.1, 13.0], p = 0.007, compared to the implantation of control biomaterials. Including only those studies with a direct comparison between control biomaterials and biomaterials loaded with biologics resulted in an improved cartilage regeneration of 14.6% [5.9, 23.4], p = 0.003. Comparing various biological cues including BMP, FGF, PRP and TGF indicated no significant differences in improvement of cartilage regeneration between these biologics.

Animal models

Evaluation of the animal models used showed no significant differences (p = 0.348) between the effects of biomaterials implanted in dogs, goats, macaques, minipigs, pigs, rabbits, rats or sheep (Table 1).

Sensitivity analyses

Sensitivity analyses were performed to assess the robustness of the meta-analysis with respect to the overall effect. The sensitivity analyses indicated that exclusion of studies with assumptions and studies written in Chinese (no risk of bias assessment analyzed) had no effect on the estimated difference in biomaterial regeneration. Moreover, including only studies with SDs or SEs in the 10–90% range did not notably change of the overall outcome effect. In a post-hoc analysis, we investigated cartilage regeneration using biomaterials implanted at different locations including condyles, femur, intercondylar fossa and the trochlea. No differences were found comparing these implant sites (p = 0.143). In another post-hoc analysis, we compared cartilage regeneration of empty defects or defects filled with biomaterials after applying microfracturing or subchondral drilling. For empty defects (p = 0.152) and biomaterial implants (p = 0.063) no significant differences between the two bone marrow stimulating techniques were found.

Publication bias

A funnel plot (Fig. 4) was prepared for all included studies to analyze the overall comparison between acellular biomaterials and non-treated empty defect controls. No extensive asymmetry was observed, indicating an absence of considerable publication bias.

Figure 4 Funnel plot of included studies to assess the overall effect of the implantation of acellular biomaterials compared to non-treated empty defect controls.

The figure indicates no substantial asymmetry.

Discussion

The regeneration of damaged cartilage has been widely investigated using preclinical models. However, the efficacy of cartilage regeneration using implantation of acellular biomaterials has never been assessed using a systematic review and meta-analysis. This systematic review aimed (a) to provide an overview of currently existing knowledge and identify knowledge gaps, (b) to provide transparency on the quality of performed in vivo studies, and (c) to aid the design of future animal studies and clinical trials. The results could provide insight in strategies for future (pre) clinical research related to biomaterial properties, incorporation of biologics, choice of a suitable animal model, and their effects on cartilage regeneration.

The general findings of this systematic review and meta-analysis are that the implantation of biomaterials improves cartilage regeneration compared to non-treated osteochondral defects by 16% (95% CI). There were only six out of 151 studies that showed a negative effect of biomaterial implantation on cartilage regeneration. In 48 studies no significant effect on cartilage regeneration was found. For those studies with improved cartilage regeneration (73 studies), clinical studies will have to confirm the beneficial effect of implantation of biomaterials on cartilage regeneration in human patients. Filardo et al. described the implantation of an osteochondral biomimetic scaffold consisting of a type I collagen cartilage-like layer, a type I collagen/hydroxyapatite intermediate layer, and a mineralized blend of type I collagen and hydroxyapatite as a subchondral bone compartment, to treat patients with osteochondritis dissecans. For these patients, clinical scores improved significantly after the first two years and evaluation by MRI indicated good defect filling and implant integration, but also heterogeneous tissue regeneration and changes of the subchondral bone (Filardo et al., 2013). In two studies included in this systematic review and meta-analysis, this osteochondral biomimetic scaffold was also implanted in sheep. Cartilage regeneration after six months was 81.8% ± 8.9% (empty defect: 23.2% ± 20.7%) and 81.2% ± 5.1% (empty defect: 23.4% ± 6.7%). A direct comparison between the degree of cartilage regeneration described in the preclinical studies and clinical study is not possible since no histological results were described in the clinical study. In addition, outcome measures used in preclinical studies may not predict the clinical outcome. For example, a randomized controlled clinical trial with BST-CarGel, a chitosan-based medical device, showed greater lesion filling and superior repair tissue quality compared to bone marrow stimulation after twelve months implantation, but without notable clinical differences related to pain, stiffness and physical function between both groups (Stanish et al., 2013). A remarkable observation is the difference in follow-up between the studies, which may explain the good histological scores in the preclinical studies after six months and heterogeneous tissue regeneration and changes of the subchondral bone after two years in human patients. In general, clinical studies demonstrated improved cartilage regeneration by the implantation of biomaterials after bone marrow stimulation, but there is still room for improvement regarding clinical outcome and tissue quality.

The only subgroup analysis that showed a statistically significant result between the groups was between control biomaterials and biomaterials loaded with biologics. In future clinical studies assessment of the beneficial properties of implanting biomaterials loaded with biologics is of interest, since a significant improvement of 8% (95% CI) compared to control biomaterials was found and even 14.6% when using studies that directly compared biomaterials with and without biologics. We were not able to perform analyses for the effect of the concentration or subtype of the growth factors due to the small size of these subgroups, although these factors may have a large effect on the outcome. In the study by Ishii et al. (2007) a positive effect of FGF-2 was observed by the addition of at least 183 ng to the biomaterials, while Maehara et al. (2010) showed significant improvements of impregnating biomaterials in 10 µg/ml and not for 100 µg/ml FGF-2. Loading biomaterials with different BMPs including BMP-2 (Aulin et al., 2013; Reyes et al., 2012; Reyes et al., 2014; Reyes et al., 2013; Tamai et al., 2005) and BMP-7 (Mori et al., 2013), or TGF subtypes including TGF-β (Mierisch et al., 2002) and TGF-β1 (Reyes et al., 2012; Reyes et al., 2014), resulted in significantly improved cartilage regeneration. However, for clinical application of these medical devices, one should take safety of the products into account as side effects of TGF-β in a joint environment, including fibrosis and osteophyte formation, have been described (Blaney Davidson, Van der Kraan & Van den Berg, 2007) and patients suffered from major complications after spinal surgery and implantation of high concentrations of BMP/INFUSE (Epstein, 2013).

The study characteristics of all included studies were tabulated to provide an extensive overview of the available literature. Besides the internal validity of the studies, the generalizability (external validity) of the study results is of great importance. The latter is affected by factors related to the animal model (species, strain, weight, age, and sex), surgery (location and size of the defect) and follow-up, resulting in heterogeneity between studies. This was also indicated by the relatively high level of heterogeneity (I2) for the main meta-analysis (99.4% [99.4, 99.4]), and the heterogeneity was almost similar for subgroup analyses. We chose to include only healthy animals receiving biomaterials. The screened studies also contained osteoarthritis models that were not included, which may be relevant for future applications to treat patients with osteoarthritis. Therefore, results from this systematic review and meta-analysis may be different compared to results found for osteoarthritis models and future clinical studies with osteoarthritis patients. We assumed that in order to assess the effect of implanted biomaterials on cartilage regeneration, reduction of the influence of confounding parameters would aid the validity of the results and conclusions. In this study, the meta-analysis included all available data of the effect of implanting biomaterials after applying bone marrow stimulating techniques (microfracture and subchondral drilling) compared to empty defects on cartilage regeneration. During microfracture surgery the subchondral bone is penetrated using an arthroscopic awl, whereas during subchondral drilling the trabecular bone is penetrated using a high speed drill, which may result in thermal necrosis (Falah et al., 2010). Remarkably, more studies applied subchondral drilling (258 studies) compared to microfracture surgery (25 studies), while microfracture surgery was developed to overcome problems associated with thermal necrosis from subchondral drilling in the treatment of human patients (Kane et al., 2013). We did perform a post-hoc meta-analysis to investigate differences in cartilage regeneration after applying both marrow stimulating techniques and subsequent implantation of biomaterials, which resulted in no significant differences between microfracturing and subchondral drilling. A reason for the larger number of animal studies performing subchondral drilling compared to microfracture surgery may be the ease to perform subchondral drilling over microfracture surgery in animals. Although in the included studies various implant locations (i.e., trochlea and condyles) were used, we grouped the results in the meta-analysis. A post-hoc subgroup analysis was performed to compare defect locations, but no overall significant differences were found for biomaterials implanted at different implant locations. Our analysis did not confirm a finding of Chen et al. (2013) showing improved chondrogenesis in trochlear versus condylar cartilage defects after bone marrow stimulation in rabbits. This may be explained by various parameters affecting the degree of cartilage regeneration at different implant locations, such as the animal model, follow-up period and rehabilitation protocol.

Different outcome measures such as macroscopic and histological evaluation, semi-quantitative macroscopical and histological evaluation using scoring systems, histomorphometry, PCR and biochemical assays were used to assess the regenerative potential of implanting biomaterials. In this systematic review and meta-analysis, only data from semi-quantitative histological scoring systems were used as outcome measure. We chose to use these data as most authors presented their results by this method and it allows quantitative comparison of different studies in a meta-analysis. Various histological scoring systems have been used by the authors of included studies, such as the O’Driscoll, Pineda, Wakitani and ICRS scoring system, which were also reviewed by Rutgers et al. (2010). Depending on the histological scoring system, parameters such as cell morphology, matrix staining, surface regularity, structural integrity, defect filling and the restoration of the subchondral bone were evaluated. A limitation of this outcome measure is that the specific topics addressed in the scoring systems greatly differ, i.e., some studies focus on the regeneration of cartilage only, cartilage as well as subchondral bone, or include a biomaterial component (e.g., scoring degradation of the implant). Other outcome measures including macroscopic evaluation, biochemical analysis and biomechanical aspects of the tissue may complete the overview of the tissue quality and provide valuable insights in articular cartilage regeneration, but these outcome measures were only used in a limited number of studies, and therefore not assessed in this analysis.

The risk of bias assessment provided insights in the quality of the experimental design of the studies. Most studies scored a low or unknown risk of bias, however, also little high risk of bias was scored. Low methodological quality (internal validity) may result in an overestimation or underestimation of the intervention effect (Higgins et al., 2011). In general, details regarding the randomization procedure were not described. Moreover, an observation during the risk of bias assessment was that only few studies included in the systematic review described that power calculations were performed, which is a crucial aspect in conducting experimental studies to ensure sufficient power of experimental designs. As a consequence, studies may lack sufficient power and thereby run the risk of false negative results. Due to the poor reporting of the experimental design for the majority of the studies the assessment of the adequacy of randomization and power calculations, and thus the assessment of the actual risk of selection bias, was inadequate. However, it may also hold true that studies were well designed but there was only poor reporting of the experimental designs (Hooijmans et al., 2012). Most researchers scoring the histology sections were blinded and sections were randomized. However, when biomaterials are not (completely) degraded, blinding between biomaterials and empty defects is practically impossible. A lack of blinding of outcome assessors implies the risk of detection/observer bias (Bello et al., 2014). Bias may have been introduced by the lack of blinding and randomization and detracts from the overall validity of the results (Bebarta, Luyten & Heard, 2003; Hirst et al., 2014). There is a risk that the positive results found are an overestimation of the true effect of using biomaterials. Introducing standardized protocols such as the golden standard publication checklist (Hooijmans et al., 2011) or the ARRIVE guidelines (Kilkenny et al., 2012) may improve reporting of animal studies.

Funnel plots represent the precision of the measured effects, which increases by an increase in study size. Therefore, for small and large studies scatter will be relatively large and little, respectively. As a consequence, generally, in the absence of bias the plot resembles a symmetrical pyramid (a funnel) (Higgins & Green, 2011). An important limitation may be publication bias, since multiple studies were included from the same author and negative results may not be published. It was described in a study by ter Riet et al. that researchers themselves estimate that only 50% of the conducted animal experiments are published. This problem may be solved by statistical corrections for publication bias (ter Riet et al., 2012). In our study, the funnel plot did not show asymmetry and therefore did not indicate the presence of publication bias.

The translational value of animal studies depends on the comparability to the clinical situation. One of the limitations of the performed animal experiments is the short follow-up times. The maximum follow-up time was one year, but most studies investigated cartilage regeneration up to six months. This limits the translational value since clinical improvements in humans are generally observed up to 1.5–3 years after microfracture surgery (Hoemann et al., 2010; Van der Linden et al., 2013). Moreover, many variations were present in the applied animal models, i.e., animal characteristics (species, strain, sex, age, weight), surgical defects (size, depth and location), applied biomaterials, and incorporated biologics. A review by Chu, Szczodry & Bruno (2010) extensively reflects on benefits and limitations of different animal models used in cartilage repair studies. They state that for humans the volume of a cartilage defect is approximately 550 mm3 and treatment is required for defects with a surface larger than 10 mm2. Due to the limited joint size of many animals, larger animal models such as minipig, goat and horse therefore offer superior translational value than smaller animals such as rats, rabbits and dogs. However, all studies contained defect volumes smaller than 550 mm3 and only few studies had defects surfaces larger than 10 mm2. Additionally, cartilage thickness differs among various species, with goat, rabbit, minipig and dogs having thinner cartilage than humans. Another drawback for some animal models is the large endogenous repair potential. In humans, untreated defects show little to no regeneration while rabbits display a large regenerative potential, limiting clinical translation. Dog, goat, minipig and horse do not have this large endogenous repair and the use of these animals may therefore be favorable. The maturity of the animals is of great importance when designing animal experiments since open growth plates can impede with the applied treatment. Animal species are skeletally mature at different ages; i.e., rabbits at the age of 16–39 weeks, pigs at 42–52 weeks, dogs at 12–24 months, sheep and goat at 24–36 months and horses at 60–72 months (Ahern et al., 2009; Chu, Szczodry & Bruno, 2010). In this study we did not group studies based on animal maturity. In addition to clinical relevance, other reasons to select an animal model are related to logistical, financial, and ethical considerations. A systematic review conducted by Ahern et al. (2009) investigated the strengths and shortcomings of different animal models and compared these with common clinical lesions in clinical studies. They remarked that smaller animal models are often used due to feasibility, while large animal models may more closely resemble humans. However, no differences were found between animal models in this systematic review and meta-analysis, which may be explained by various parameters affecting the degree of cartilage regeneration such as implant location, defect size, follow-up period and rehabilitation protocol.

In this systematic review and meta-analysis the efficacy of cartilage regeneration using acellular biomaterials was compared to the natural healing response of defects treated with microfracture surgery and subchondral drilling. The risk of bias assessment indicated poor reporting in animal studies, which may be improved in future animal studies. Moreover, to improve the translation towards clinical trials animal experiments should be comparable to the clinical situation. As described in this systematic review a relatively high level of heterogeneity exists between studies related to the animal model, surgery and follow-up, with a need to resemble current clinical settings more closely. In this study we only addressed bone marrow stimulating techniques (microfracture and subchondral drilling) and subsequently the incorporation of biomaterials, but also the regeneration of partial thickness cartilage defects may be beneficial to prevent progression to full-thickness cartilage defects, limit the progression towards osteoarthritis and improve quality of life in patients. In many studies also cell-laden biomaterials have been implanted and the beneficial effect of cellular biomaterials versus acellular biomaterials and the natural healing response has been studied. Although acellular biomaterials offer various advantageous properties over cellular biomaterials such as no donor-site, no cell culture, off the shelf availability, less regulatory issues, and application of one-stage surgical procedures (Brouwer et al., 2011; Efe et al., 2012), studying the additive value of cellular biomaterials may aid further improvement of marrow stimulating techniques.

Conclusion

The systematic review and meta-analysis resulted in a structured, thorough and transparent overview of literature related to the current evidence for the efficacy of cartilage regeneration using acellular biomaterials implanted after microfracturing in animal models. Cartilage regeneration is more effective by implantation of acellular biomaterials in microfracture defects compared to microfracturing alone. The efficacy is further improved by the incorporation of biologics.

Supplemental Information

Supplemental Information 1 Search strategies

Click here for additional data file.

Supplemental Information 2 Methodological quality assessment tool

Click here for additional data file.

Supplemental Information 3 Study characteristics

Click here for additional data file.

Supplemental Information 4 Study characteristics; abbreviations and references

Click here for additional data file.

Supplemental Information 5 Raw data

Click here for additional data file.

Supplemental Information 6 Risk of bias results per individual study

Risk of bias results per individual study

Click here for additional data file.

Supplemental Information 7 Forest plot

Click here for additional data file.

Supplemental Information 8 Prisma checklist

Click here for additional data file.

We thank Jie An (Department of Biomaterials, Radboud Institute for Molecular Life Sciences, Radboud university medical center) for full-text screening articles written in Chinese. Gerrie Hermkens from the Radboud university medical center medical library is greatly acknowledged for help retrieving full text studies.

Additional Information and Declarations

Competing Interests

Author Contributions

Data Availability

The authors declare there are no competing interests.

Michiel W. Pot conceived and designed the experiments, performed the experiments, analyzed the data, contributed reagents/materials/analysis tools, wrote the paper, prepared figures and/or tables, reviewed drafts of the paper.

Veronica K. Gonzales performed the experiments, reviewed drafts of the paper.

Pieter Buma reviewed drafts of the paper.

Joanna IntHout analyzed the data, contributed reagents/materials/analysis tools, prepared figures and/or tables, reviewed drafts of the paper.

Toin H. van Kuppevelt conceived and designed the experiments, reviewed drafts of the paper.

Rob B.M. de Vries conceived and designed the experiments, analyzed the data, contributed reagents/materials/analysis tools, reviewed drafts of the paper.

Willeke F. Daamen conceived and designed the experiments, analyzed the data, reviewed drafts of the paper.

The following information was supplied regarding data availability:

The raw data has been supplied as Data S1.

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
