# Peer review of "Improved cartilage regeneration by implantation of acellular biomaterials after bone marrow stimulation: a systematic review and meta-analysis of animal studies"

_PeerJ, doi:10.7717/peerj.2243_

## Round 0.1 · original submission · Minor Revisions

The referees have provided a number of constructive comments to strengthen the manuscript. Could you please address these in your revision?

Reviewer 1 ·

Basic reporting

The review submitted by Pot and colleagues provides a meta-analysis about the effect of acellular biomaterials after bone marrow stimulation in animal studies in regards to cartilage regeneration Interestingly the authors could demonstrate the implantation of acellular biomaterials significantly improves the cartilage regeneration compare to non-treated controls and that the addition of biologicals further improves the cartilage regeneration compared to control biomaterials. The aim of the review is not only to provide insights about the present research, but also to identify the knowledge gaps and aid the design of future animal experiments and clinical trials.

- The text:
It is clear and well written. A few remarks:
Line 113: the use of the bracket can be changed [(
Line 627: “.” Instead of “:”
Line 415: the sentence is not clear
- Intro and background:
Remark 1: The difference between microfracture and subchondral drilling is only described in the final part of the review. It would be useful to define the two procedures in the introduction and comment on the low sample size for microfracture (25 studies against the 258 for subchondral drilling)
Remark 2: a definition of “biologics” should be given in the introduction
Remark 3: rats and horses are mentioned in the materials and methods but not further in the results and discussion (line 273)
- Literature:
Line 867: missing part of the reference
- Figures/ Tables
Possible improvements for the Table 1: specify what studies and groups are. PRP is not a growth factor, it can be addressed as “biologics”. Why there is no CI in the animal models category? “studies” and “groups” are not defined.
Validity of the findings
- impact and novely:

Experimental design

We believe that by covering 283 studies, the review gives a useful and complete overview of the existing knowledge on the topic until April 2015.
Question 1: Is the publication year a criterion? (line 154)
Line 445: how did you define tissue quality?
Line 381: PRP is not a growth factor

Validity of the findings

We believe the goals stated in the introduction are achieved. We appreciate that this review can identify the issues of the field, notably a lack of standardization which triggers a high level of heterogeneity between the studies. Indeed, the investigated studies differ in the animal model characteristics (age, sex, species…), in the surgery type (microfracture, subchondral drilling, surgical procedures), in the location and size of the defects, in the time points and in the histological scoring methods. It is thus very interesting to see that information can be withdrawn from such a variety of studies.
Nevertheless we believe that the review could better fulfill the last aim and effectively help the design of future experiments by providing the cartilage regeneration score in the Supplementary information 3a of each study. This would allow the reader to effectively track the outcome of the single papers.

Reviewer 2 ·

Basic reporting

The article has sufficient introduction, background and referencing. The structure of the article is appropriate for a meta-analysis and the submission is self-contained. All appropriate supplemental data is supplied. The manuscript could benefit from restructuring of certain sentences so that they were more easily understood, particularly the first few lines of the abstract, lines 62-63, 101-103, and 322.

Experimental design

The experimental design and steps of conducting the meta-analysis and risk assessment are quite clearly defined. However, there is some confusion based on whether subcondral bone quality and regeneration was evaluated in the scoring of cartilage regeneration, and if so, whether the meta-analysis is truly a measure of cartilage regeneration only. The following comments would further aid in fully understanding the experimental design and results.

1. In results 3.2, the authors state subchondral drilling was performed in 258 studies compared to microfracture surgery in only 25 studies. This implies that the majority of studies were osteochondral biomaterials aimed at restoring both chondro and osteo tissues, however the study stresses throughout that cartilage regeneration was assessed. In the meta-analysis, were the results relating to bone regeneration ignored or were improvements in bone quality/restoration incorporated in scoring? Since the majority of studies in this analysis were performed with subcondral drilling, it might be more accurate to report degree of osteochondral regeneration instead of cartilage regeneration alone.

2. For the meta-analysis it is stated that semi-quantitative histological marks were used to score cartilage regeneration however no details on the general histological data used to create this score is provided. Was scoring based primarily on degree of GAG staining, collagen staining, immunohistochemistry, macrophage staining, cell morphology, cell infiltration, damage to surrounding cartilage, ect. I realize this was different for all the studies and was commented on some in the discussion; however a general idea of the measures that contributed to the scoring in the methods or results would further inform/support the meta-analysis findings. Further, related to question 1, was degree of subchondral bone restoration and osteogenic markers included? If so, is the scoring of cartilage regeneration truly cartilage regeneration, or more accurately described as osteochondral regeneration?

3. It would be helpful for the authors to clearly define what the risk of bias analysis is for readers unfamiliar with this method. Clearly defining that it analyses the studies of the review, and not the review itself, along with simplifying the questions asked to more general terms such as randomization, power analysis, implant/analysis blinding, ect would be very helpful. Further the authors switch between calling it methodological quality assessment and risk of bias analysis. Specifically, the sections in the methods and results are called methodological quality assessment, but in Figure 2 it is referred to as Risk of bias assessment. It would be helpful to stick with one term to link the sections and the figure better.

4. In Figure 2, what do authors mean by “no primary study” as a means of exclusion? Also, authors exclude studies based on no osteochondral defect, however throughout the article it is stated that cartilage regeneration with bone marrow stimulation is being assessed. In the case of microfracture or abrasion, it would be possible to have a full thickness cartilage defect. I assume all included studies were osteochondral defects and full thickness defects, and only partial thickness defects were excluded. Please clarify.

Validity of the findings

Findings are valid, and the authors can clearly say the presence of a biomaterial improves cartilage regenerations and the addition of a biological cue further improves the regeneration. However, the authors should be cautious in saying that there is no effect from the other subgroups analysed. Many of these other subgroups depend on multiple factors to have an effect and the number of studies performed becomes limited, thus the grouping of the meta-analysis might be too general or limited to pick up effects from animal model, defect location, follow up time, and defect size. Specifically related comments are below.

1. In the 4th paragraph of the discussion, lines 491-493, the authors state their analysis did not confirm a finding of Chen at al showing improved chondrogenesis in trochlear versus chondylar cartilage. I would assume degree of healing in different implant locations would depend greatly on animal model used, follow up time, and rehab protocol (ie intermittent active motion, continuous passive motion, or immobilization) which would have major effects on the loading environment of the implant. It seems that in order to make a conclusion on the effect of implant location further subgrouping into animal model, follow up time, and rehab protocol would be needed and in this case the power of the analysis would most likely become too low. Further, in line 566 the authors state no differences were found between animal models in the review. Degree of healing in particular animal models is most likely affected by implant location, defect size and rehab routine. Some discussion on the multiple factors effecting degree of healing in implant locations and animal models and any trends observed in the data/study setups would be further informative.

2. In the discussion the authors comment some on the effect of defect size. Defect size should have a major effect on the ultimate success of an implant and is most like one of the major contributors to inconsistencies between in vivo studies. It would be interesting to expand the meta-analysis to defect size with sub-populations of animal model to account for size of the joint (or large animal vs small animal), or just within the rabbit model since it appears most studies have been performed in this model. Possibly this could stress the importance of moving toward critical sized defects or defect sizes that correlate to human defect sizes.

Additional comments

1. Throughout the paper the authors are very general with the use of the word “biomaterials” leading to a lack of clarity in the introduction. In the 3rd paragraph of the introduction the authors say “The construction of implants, so called biomaterials..” Generally biomaterials make up implants, but implants are not only biomaterials. Often implants have biomaterials constructed in some way to match the geometry of the replacement site or the biomaterials have been manipulated in some way to elicit a certain response, such as electrospinning, foaming, directional freezing, application of biologics, ect. Further the authors say, “to combine the advantageous properties of these materials, multi-layered biomaterials have been constructs.” For ease of understanding I would suggest being more specific between biomaterials vs construct/implant. Biomaterials can be synthetic or natural, and formed as a dry scaffold or hydrogel, however once you start getting to layering of different biomaterials, and different processing of biomaterials, the biomaterials become part of a construct/implant.

2. In Figure 1 and throughout the paper authors refer to biological cues as being only growth factors for chondrogenic and osteogenic differentiation. While this is primarily true for chondrogenic stimulation, osteogenic biologics include other triggers such as mineralization, hydroxyapatite, or β-glycerophosphate. Further, later in the discussion and in the supplemental material it is noted PRP is considered a biological cue. It would be helpful to expand the discussion of biological cues in the introduction, figure 1, and results to include examples of cues beyond growth factors.

3. Why did the authors choose to eliminate osteoarthritic animal models, when this model is most likely more relevant to treatment options than a healthy animal? It would be interesting to compare biomaterial effect in osteoarthritic animal models vs healthy animal models.

4. In section 2.2 Study selection, lines 150-153, the authors state “articles were excluded”, but then in the sentence after say “articles were included”. This seems to imply that no studies were actually excluded at this step. Please clarify.

5. In results 3.2, authors state “generally, the animals were older in large animal models compared to animals used in small animal models.” Is this statement relating more to the life expectancy/cycles of animals than trends in research? A small animal model carried out on 6 month old animals could be an adult and at a similar developmental point as that of a large animal model which is >2 years old. It would be interesting to have some discussion on the effect of age of maturity (neonatal, adolescent, adult) on regeneration of cartilage with biomaterials and the need for studies to report the maturity of models for better comparison between results of studies.

6. Figure 3 is somewhat confusing in presentation. On first look it appeared the unknown risk included the low risk also, and that the high risk included low risk and unknown. A rewording of the x-axis or further clarification in the caption stressing it is depicting the percentage of each kind of risk within the entire population would be helpful.

7. In section 3.4.1 could authors please clarify lines 356-358. Did the presence of a biomaterial significantly improve regeneration over empty defects at short term, long term, and maximum follow-up times?

---

## Round 0.2 · accepted · Accept

Thank you for providing a clear outline of the changes you made to the manuscript based on the referees' comments.